# Impact of Treatment with Ustekinumab on Severe Infections in a Patient with Uncontrolled Psoriasis and Late-Onset Combined Primary Immunodeficiency: Case Report

**DOI:** 10.3390/pathogens12091156

**Published:** 2023-09-12

**Authors:** Luiz Euribel Prestes-Carneiro, Marilda Aparecida Milanez Morgado de Abreu, Eduardo Vinicius Mendes Roncada, Diego Garcia Muchon, Fernanda Miranda Caliani, Dewton Moraes Vasconcelos

**Affiliations:** 1Immunodeficiencies and Infectious Diseases Outpatient Clinic, Faculty of Medicine, Oeste Paulista University, Presidente Prudente 19.050-920, Brazil; diego.muchon@gmail.com (D.G.M.); fernandacaliani@hotmail.com (F.M.C.); 2Department of Dermatology, Faculty of Medicine, Oeste Paulista University, Presidente Prudente 19.050-920, Brazil; marilda@morgadoeabreu.com.br (M.A.M.M.d.A.); eduardoroncada.dermatologia@hotmail.com (E.V.M.R.); 3Laboratory of Medical Investigation Unit 56, Hospital das Clínicas da Faculdade de Medicina da Universidade de São Paulo (HC-FMUSP), São Paulo 05403 000, Brazil; dewton.vasconcelos@hc.fm.usp.br

**Keywords:** late-onset combined immunodeficiency, psoriasis, ustekinumab, infections

## Abstract

A 35-year-old man with a late-onset combined immunodeficiency (LOCID) variant of common variable immunodeficiency, severe plaque psoriasis, psoriatic arthritis, and Crohn’s disease was attended in the Regional Hospital of Presidente Prudente and HC-FMUSP, São Paulo, Brazil. Anti-IL-12/IL-23 (ustekinumab) monoclonal antibody was prescribed due to the failure of other treatments (phototherapy, oral acitretin) for psoriasis and a Psoriasis Area Severity Index >10. We evaluated the impact of treatment with ustekinumab on severe infectious diseases in a patient with uncontrolled psoriasis and LOCID followed for 8 years. Four quarterly doses of ustekinumab 90 mg and human immunoglobulin replacement (10,000 mg at 28-day intervals) were administered. Immunophenotyping, cultures of lymphocytes, genetic sequencing, and whole exome sequencing were performed to investigate the primary immunodeficiency. Normal lymphocyte proliferation; pathogenic variants in genetic sequencing, and clinically significant variants in the whole exome for primary immunodeficiencies were not detected. The main infections before and after treatment with ustekinumab were chronic sinusitis and gastroenteritis. The patient was infected with COVID-19, dengue (twice) and influenza and was hospitalized three times for intravenous antibiotic therapy. Ustekinumab did not influence the susceptibility of the patient with LOCID to severe infections and significantly improved psoriasis, psoriatic arthritis, and Crohn’s disease.

## 1. Introduction

Primary immunodeficiencies or innate inborn errors of immunity (IEI) are a heterogeneous group of diseases that result from defects in the development and/or function of the immune system. They are classified as disorders of adaptive immunity (i.e., T cell, B cell, or combined immunodeficiencies) or innate immunity (e.g., phagocyte and complement disorders). Although the clinical manifestations are variable, most disorders at least involve increased susceptibility to infection, autoimmunity, and tumors, and about 60% of these patients have decreased antibody levels [1]. IEI should be suspected in patients with recurrent upper airway infections (otitis or sinusitis), recurrent pneumonia, continuous use of antibiotics, skin abscesses or atopic dermatitis; or a family history of IEI [1].

Autoimmune diseases are a group of polygenic, immune-mediated genetic diseases and may involve any organ or system. Although the pathophysiology is still unclear, in all cases, autoantibodies formed against the individual’s own tissue begin to destroy it, including the nervous, digestive and respiratory systems and the skin, blood, eyes, joints, and endocrine glands [2,3,4]. Autoimmune manifestations are observed with considerable frequency in patients with primary antibody deficiencies, including common variable immunodeficiency and selective IgA deficiency, but they can also be evident in patients with combined immunodeficiency disorders [3,4]. Supporting the association between autoimmune diseases and IEI, based on the concept that excessive inflammatory responses and autoimmunity are also common manifestations of immunodeficiency, the overall prevalence of autoimmunity in patients with CVID was 29.8%, and the prevalence of autoimmune hematologic diseases, autoimmune gastrointestinal disorders, autoimmune rheumatologic disorders, autoimmune skin disorders, and autoimmune endocrinopathy was 18.9%, 11.5%, 6.4%, 5.9%, and 2.5%, respectively [4].

Cutaneous manifestations are common in patients with IEI and therapies include topical agents, phototherapy, and immunosuppressive or immunobiological agents [5]. Over the past 2 decades, several immunobiological agents have been developed for the treatment of moderate-to-severe plaque psoriasis. The first class included the TNF-α inhibitors and the monoclonal antibodies adalimumab and infliximab, followed by the monoclonal antibody ustekinumab, which is an IL-12/23 inhibitor. A newer class of monoclonal antibodies is the IL-17 inhibitors, including secukinumab and ixekizumab, which block IL-17A, as well as brodalumab, which blocks the IL-17 receptor (IL-17RA). Finally, the IL-23 inhibitors guselkumab, tildraquizumab and risankizumab were added [6]. The role of ustekinumab in triggering infections in immunocompetent patients treated for psoriasis and Crohn’s disease has been investigated [7,8]. However, data on patients with IEI with combined immunodeficiency are scarce because these patients could not be included in clinical studies. Our objective was to investigate the impact of treatment with ustekinumab in triggering infections in a patient with severe uncontrolled psoriasis, psoriatic arthritis, Crohn’s disease, and late-onset combined immunodeficiency (LOCID) over an 8-year period.

## 2. Case Report

In 2016, a 33-year-old man was referred to the dermatology and immunodeficiency outpatient clinic at Presidente Prudente Regional Hospital (RH), São Paulo state, Brazil. Skin disorders were present in our patient soon after birth in our patient. At 3 months of age, he had atopic dermatitis and he was 1 year old when he was first hospitalized due to a severe infection in the lower limb after an insect bite. Subsequently, throughout his childhood, he was hospitalized about five times a year due to mild and severe skin infections, including cellulitis and erysipelas, in addition to ulcerative atopic dermatitis, making it clear that skin lesions would be a major health problem in his life.

In 2006, at the age of 23 years, he was diagnosed with LOCID and severe psoriasis and referred from Presidente Prudente to the primary immunodeficiency outpatient clinic at the Faculty of Medicine of the University of São Paulo, São Paulo (HC). Due to hypogammaglobulinemia, immunoglobulin deficiency, and alterations in immunophenotyping with low levels of CD3/CD4 and CD19 cells (Table 1), he was given intravenous human immunoglobulin (IVIG) replacement at a dose of 400 mg/kg at 28-day intervals, and he showed significant improvement. In 2011, IVIG replacement was discontinued due to severe anaphylaxis and he underwent follow-up at HC. From 2011 to 2016, without immunoglobulin replacement, his health condition, psoriasis, and infections were exacerbated.

In May 2016, at the first consultation in RH and after 5 years without follow-up in a specialized immunodeficiency service and without replacement of IVIG, his state of health was poor, with multiple infections and comorbidities, mainly in the skin, and worsening of his psoriasis, leading to difficulty carrying out basic daily activities. In November 2016, subcutaneous immunoglobulin replacement was started, which was a new form of therapy that protected the patient from an anaphylactic reaction and adverse events. He had widespread pruritic erythematous scaly plaques, including the palmoplantar regions (Figure 1), which were not controlled after using topical corticosteroids, systemic acitretin, and UVB-narrow band phototherapy. The Psoriasis Area Severity Index was >10. Tests, including serology for hepatitis B and C, HIV and syphilis, PPD (purified protein derivative), and chest radiographs were requested before starting immunobiological therapy. Given the importance of his immunodeficiency, it was decided not to give him methotrexate or anti-TNF-α agents and ustekinumab was indicated.

Before starting treatment in this immunosuppressed patient, prophylaxis with isoniazid for *Mycobacterium tuberculosis*, sulfamethoxazole-trimethoprim for pneumocystosis, fluconazole for candidiasis, and azithromycin for atypical mycobacteria were given. About 30 days after the use of the prophylactic drugs, he had important side effects, including diarrhea, epigastric pain, dehydration, and a decline in his general condition requiring hospitalization, and prophylactic treatment was interrupted. On 14 February 2017, the standard initial dose of ustekinumab (45 mg at weeks 0 and 4) was administered subcutaneously followed by a maintenance dose of 45 mg every 3 months, which is the standard label dosage for psoriasis with or without arthritis for adults weighing < 100 kg.

He showed significant improvement after 2 doses of medication (Figure 2) and remained on this dosage until the end of 2019. In 2020, due to the worsening of his joint condition, and the diagnosis of psoriatic arthritis, the dose of ustekinumab was adjusted. First, it was doubled (90 mg every 3 months), but the joint pain did not improve; then, the interval between doses was decreased (90 mg every 8 weeks).

In May 2022, the patient presented with worsening of abdominal pain and diarrhea, leading to dehydration and weight loss. Colonoscopy with biopsies revealed Crohn’s disease and upper digestive endoscopy with a duodenal biopsy that demonstrated villous atrophy and infiltration of intraepithelial lymphocytes, compatible with celiac disease; there was clinical improvement of diarrhea after suspension of gluten intake, this led to improvement in Crohn’s disease and psoriasis.

From 2017 to 2022, he was infected with COVID-19, dengue (twice), and influenza virus, and was hospitalized three times for intravenous antibiotic therapy. The patient was infected with the dengue virus in 2018 and with dengue, COVID-19 and influenza viruses in 2022. Dengue infection is endemic in Presidente Prudente and in 2022, 17,493 individuals were infected. As with dengue, in December 2022, 1:3.2 inhabitants were infected with COVID-19 [9]. The patient’s wife and daughter were also infected with dengue and COVID-19 at the same time. He was infected with influenza virus by his parents, whom he helped care for. We suggest that treatment with ustekinumab is not responsible for the infection caused by these viruses.

At present, he continues with the optimized treatment with ustekinumab and replacement of subcutaneous immunoglobulin every 28 days. He is being followed at immunology, dermatology, rheumatology, and gastroenterology clinics.

With regard to family history, the sister is being followed due to juvenile idiopathic arthritis and vitiligo; his brother has rhinitis, chronic sinusitis, diarrhea, anal fistula, and bilateral deafness after using monoclonal antibodies to treat the anal fistula; a cousin was diagnosed with common variable immunodeficiency, thrombocytopenia, and autoimmune hemolytic anemia (died aged 28 years); an aunt was diagnosed with combined immunodeficiency, thrombocytopenia and autoimmune hemolytic anemia (died aged 54 years); a nephew died soon after birth due to probable severe combined immunodeficiency syndrome.

In 2016, ustekinumab was approved in Brazil for the treatment of psoriasis in the case of failure or contraindication of a classic systemic treatment (in his case, phototherapy and acitretin).

### Genetic Testing

Genetic sequencing for primary immunodeficiencies was performed through exon capture (peripheral blood) with Nextera Rapid Capture Mendelics Custom Panel V2, followed by next-generation sequencing with Illumina HiSeq (alignment and identification of variants using bioinformatics protocols with reference to the GRCh37 version of the human genome) (Mendelicx, São Paulo, Brazil). The following genes were analyzed: ADA, AICDA, BLNK, BTK, CASP10, CASP8, CD19, CD247, CD3D, CD3E, CD3G, CD40, CD40LG, CD79A, CD79B, CD8A, CIITA, CYBA, CYBB, DCLRE1C, ELANE, FAS, FASLG, FOXN1, FOXP3, G6PC3, GATA2, GFI1, HAX1, IFNGR1, IFNGR2, IGLL1, IL12RB1, IL. Pathogenic variants of 2RG, IL7R, JAK3, LIG4, LRRC8A, LYST, MAGT1, MPO, MYD88, NCF1, NCF2, NCF4, NFKBIA, NHEJ1, NRAS, ORAI1, PIK3CD, PNP, PRF1, PTPRC, RAB27A, RAC2, RAG1, RAG2, RFX5, RFXANK, RFXAP, SERPING1, SH2D1A, STAT1, STAT5B, STX11, STXBP2, TAP1, TAP2, TAPBP, UNC13D, UNG, WAS, WIPF1, XIAP were not found.

Whole-exome sequencing was performed using 3B-Exome Report and no clinically significant variant was detected (3 Billion, Republic of Korea). The implications of inconclusive genetic tests on the patient lead us to expand the study through more precise techniques and to re-analyze the results of genetic sequencing.

## 3. Discussion

The implication of inconclusive molecular analysis did not result in a diagnosis and the patient’s immunodeficiency was not explained by an underlying genetic alteration. The lack of a genetic diagnosis, given his extensive family history, suggests a hereditary combined immunodeficiency phenotype. Variable defects of cellular and humoral immunodeficiency were found in the patient and his family, suggesting a hereditary combined immunodeficiency phenotype. Specific combined hereditary immunodeficiencies are rare diseases, but they are increasing as more single-gene diseases are genetically defined (Figure 3).

The patient was 16 years of age when he developed severe plaque psoriasis and he was referred from Presidente Prudente, in the countryside, to a reference center for immunodeficiency in São Paulo (HC), with a diagnosis of LOCID. Despite presenting alarming signs of IEI, including persistent severe infections of the upper and lower airways, multiple annual hospitalizations for infectious diseases and autoimmune disorders, IEI was diagnosed late in this patient. The distribution of IEI reference centers in Brazil is poor; IEI network centers are concentrated mainly in big cities and capitals of the southeast region [10]. In Presidente Prudente, the outpatient immunodeficiency clinic was established only in 2014, and on that occasion, more than 12 patients diagnosed with IEI were unaccompanied, without monthly immunoglobulin replacement, and mostly in a poor state of health, similar to our patient [10].

It is well known that anti-TNF-α agents increase the risk of re-activation or triggering of infectious diseases compared with anti-interleukin agents [11]. Due to our patient’s diagnosis of combined immunodeficiency, the ideal treatment was based on a drug that could treat psoriasis in all its manifestations with a highly favorable safety profile. Ustekinumab has a low risk of infections compared with anti-TNF-α agents [11,12]. Furthermore, ustekinumab was the only anti-interleukin agent available and approved in the public health system in Brazil at the time the treatment was started. Although the efficiency of ustekinumab in treating psoriatic arthritis is lower and slower compared with anti-TNF-α agents, ustekinumab was able to contain joint and skin disease for a long time [12]. According to the World Health Organization, about 25% of people are infected by the tuberculosis bacillus worldwide; 1 in every 4 people are infected in Brazil. Our patient was diagnosed with a primary combined immunodeficiency with low levels of CD3/CD4 and CD19. The use of immunosuppressants such as TNF inhibitors, corticosteroids and other immunosuppressants in patients infected with latent *M. tuberculosis* increases the risk of progression to active tuberculosis [13]. In Brazil, patients in this risk group undergo prophylactic treatment with isoniazid before or concomitantly with treatment with immunosuppressants. However, no guidelines were found for prophylactic treatment for primary combined immunosuppressed patients treated with ustekinumab.

For our patient, treatment with ustekinumab and the possibility of monthly subcutaneous replacement with immunoglobulin was a turning point in his life, with improvements in his skin psoriasis, psoriatic arthritis, and Crohn’s disease as well as infectious disease control. Furthermore, the beneficial effect that immunoglobulin replacement has on autoimmune diseases is well documented, decreasing their intensity and modulating their activity [14,15]. In 2017, when he started ustekinumab, a broad range of literature was available on its effects in triggering infectious diseases in healthy individuals [7,8,16,17], however, as far as we know, no literature was available on combined primary immunodeficient patients. We were concerned that by using an immunosuppressive drug on a patient diagnosed with LOCID, his infections might become more frequent and severe, putting his life at risk. He arrived at RH in May 2016, and the spectrum of infections in 2015 was described by the patient and characterized mainly by upper airway infections, including sinusitis and tonsillitis. However, gastroenteritis, a recurrent infection commonly present after ustekinumab administration, was already present in the period 2015–2016. The patient has a combined primary immunodeficiency with low levels of CD3, CD4 and CD19, and contrary to what we expected, his history of infections between 2017 and 2022 showed a controlled number of infections which were common to patients with CIVD. Sinusitis and gastroenteritis were the most prevalent infections, with no differences before and after the use of ustekinumab. The patient was hospitalized only three times to receive high-spectrum intravenous antibiotics. The mechanisms by which ustekinumab triggers infectious diseases are well described.

For ustekinumab, multiple cases of infections have been described in clinical trials and case reports [16,17,18]. Recently, the World Health Organization Pharmacovigilance Center published the global risk of bacterial skin and Herpesviridae infections with ustekinumab, secukinumab, and TNF-α inhibitors. For bacterial skin infections, ustekinumab showed the strongest association compared with secukinumab and TNF-α inhibitors. Ustekinumab showed a higher relative risk for skin infections among patients with psoriasis [11]. However, conflicting results were found in a nationwide cohort study from France covering approximately 99% of the French population. In the association between the use of biologicals and the risk of serious infections in patients with psoriasis, ustekinumab was associated with a lower risk of having a serious infections compared with adalimumab or infliximab and etanercept [19].

It is important to emphasize that CVID is a clinical and immunological description of a syndrome characterized by a decrease in the number and function of antibodies leading to recurrent severe infections and often requiring inpatient antibiotic therapy. Due to the great communication of biological pathways, different approaches can be used for different patients with a similar clinical picture, raising the possibility of controlling immune dysregulations. They tend to be a significant aggravating factor for patients with CVID, one which is very common in the clinical practice of specialists as well as generalists around the world [20].

Because patients with primary or secondary immune deficiency cannot be included in clinical trials, few publications on the role of ustekinumab treatment in triggering infectious diseases are available. A patient with Crohn’s disease and transient IgM and IgG immunodeficiency treated with ustekinumab presented fungal endocarditis, esophageal moniliasis, and septic condition of undetermined origin [21]. In a cohort of nine patients with chronic granulomatous diseases (CGD), ustekinumab was found to be safe with efficacy for the treatment of CGD-associated inflammatory bowel disease [18].

Our study is innovative because few reports are available in the scientific literature on the role of ustekinumab in triggering infectious diseases in patients with IEI and there are no reports on patients with combined primary immunodeficiency. Furthermore, the study has global relevance because the ustekinumab treatment continues to be approved for more indications, increasing the number of patients at risk of infectious diseases. It is important to determine the risk of infectious diseases and be permanently vigilant.

## 4. Conclusions

A case report is presented of a patient with LOCID in whom treatment with ustekinumab, a immunosuppressor, did not influence susceptibility to severe infections and significantly improved psoriasis, psoriatic arthritis, and Crohn’s disease.

## Figures and Tables

**Figure 1 pathogens-12-01156-f001:**
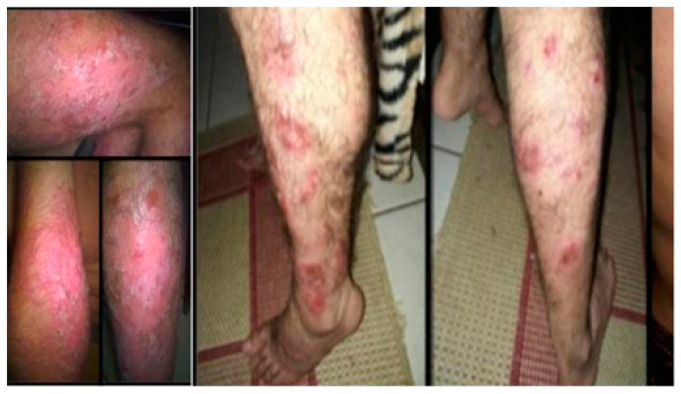
Patient with diffuse erythematous scaly plaques before ustekinumab treatment. Pictures obtained in 21 November 2016, in the Ambulatory of Dermatology of the RH.

**Figure 2 pathogens-12-01156-f002:**
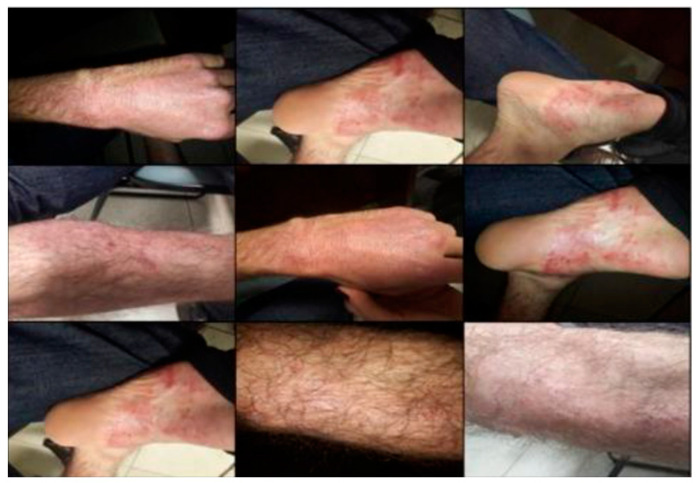
Significant improvement in psoriatic lesions after 2 applications of ustekinumab. Pictures obtained on 26 April 2017, in the Ambulatory of Dermatology of the RH.

**Figure 3 pathogens-12-01156-f003:**
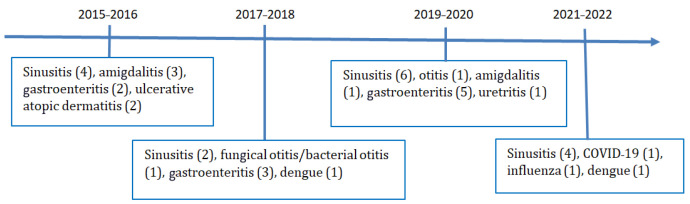
Infectious diseases before treatment (2015–2016) and after treatment with Ustekinumab (2017–2022). The number of infectious diseases in the period is shown in parentheses.

**Table 1 pathogens-12-01156-t001:** Immunological findings on diagnosis.

Immunophenotyping	Patient	Normal Range
CD45/CD3	564	605–2460 cells/µL
CD45/CD3/CD4	291	493–1666 cells/µL
CD45/CD3/CD8	235	224–1112 cells/µL
CD45/CD19	51	72–520 cells/µL
CD45/CD3^−^/CD16^+^	51	73–654 cells/µL
Lymphocyte proliferation		Stimulation Index
Phytohemagglutinin	37.9	18.28–343.00
Pokeweed	19.5	8.42–107.40
OKT3	43.4	15.62–219.30
Response to vaccination		
Anti-HBs	151.2	≥10 IU/mL
Anti-tetanus	0.95	>0.8 IU/mL
Anti-diphtheria	0.35	>0.1 IU/mL
Anti-pneumo 23		>50%
Immunoglobulins		
IgG	318	700–1600 mg/dL
IgM	24	40–230 mg/dL
IgA	96	70–400 mg/dL
IgE	<0.5	<140 mg/dL
IgG subclasses		
IgG1	5.62	4050–10,110 mg/L
IgG2	1.97	1690–7860 mg/L
IgG3	400	110–850 mg/L
IgG4	617	30–2010 mg/L

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
