# Peer review of "Impact of Treatment with Ustekinumab on Severe Infections in a Patient with Uncontrolled Psoriasis and Late-Onset Combined Primary Immunodeficiency: Case Report"

_pathogens, 2023, doi:10.3390/pathogens12091156_

Round 1
Reviewer 1 Report
The authors present a clinical case of a patient with late-onset common variable immunodeficiency associated with psoriasis/ psoriatic arthritis treated with ustekinumab combined with immunoglobulin therapy.
Some points of the reported clinical case need clarification:
1) Why treatment with ustekinumab was chosen and not with other biologics e.g., anti-TNFalpha. even considering that the effect on psoriatic arthritis was modest and required higher and more frequent dose administration of ustekinumab.
2) Why the patient received preventive anti-mycobacterium tuberculosis and antifungal therapy, considering that common variable immunodeficiency is chracterized by antibody deficiency rather than T cell deficiency.
3) The patient received three hospitalizations (COVID-19, Dengue virus infection and influenza). The claim that ustekinumab did not contribute to these hospitalizations needs to be clarified.
4) Line 127: Loading dose of what? This needs to be specified
5) Ustekinumab has immunosuppressive effects, but it is not more potent than other biolotechnology drugs used for psoriasis/ psoriatic arthritis, and it is not considered a particularly potent immunosuppressant. Therefore, the term "strong" should be eliminated.
5) It should be specified that inhibition of IL-23 acts on Th17 cells.
6) Line 245: Ustekinumab is approved for the treatment of Cronn's disease.
English needs minor editing
Author Response
REVIEWER 1
Dear Reviewer
Thank you very much for your valuable comments that certainly contribute to the improvement of our manuscript. We tried to fill in all the issues raised and amendments necessary.
1) Why treatment with ustekinumab was chosen and not with other biologics e.g., anti-TNFalpha. even considering that the effect on psoriatic arthritis was modest and required higher and more frequent dose administration of ustekinumab?
Answer: An explanation about the reasons why ustekinumab was chosen was added to the text as follows: Lines 212-220
It is well known that anti-TNF-α agents increase the risk of re-activation or triggering of infectious diseases compared with anti-interleukin agents [11]. Due to our patient's diagnosis of combined immunodeficiency, the ideal treatment was based on a drug that could treat psoriasis in all its manifestations with a highly favorable safety profile. Ustekinumab has a low risk of infections compared with anti-TNF-α agents [11,12]. Furthermore, ustekinumab was the only anti-interleukin agent available and approved in the public health system in Brazil at the time the treatment was started. Although the efficiency of ustekinumab in treating psoriatic arthritis is lower and slower compared with anti-TNF-α agents, ustekinumab was able to contain joint and skin disease for a long time [12].
2) Why the patient received preventive anti-mycobacterium tuberculosis and antifungal therapy, considering that common variable immunodeficiency is chracterized by antibody deficiency rather than T cell deficiency.
- A text was added in the manuscript Lines 221-230
According to the World Health Organization, about 25% of people are infected by the tuberculosis bacillus worldwide; 1 in every 4 people are infected in Brazil. Our patient was diagnosed with a primary combined immunodeficiency with low levels of CD3/CD4 and CD19. The use of immunosuppressants such as TNF inhibitors, corticosteroids and other immunosuppressants in patients infected with latent M. tuberculosis increases the risk of progression to active tuberculosis [13]. In Brazil, patients in this risk group undergo prophylactic treatment with isoniazid before or concomitantly with treatment with immunosuppressants. However, no guidelines were found for prophylactic treatment for primary combined immunosuppressed patients treated with ustekinumab.
The question was discussed by one of the authors, Dr. Dewton Moraes Vasconcelos, head of the Laboratory of Medical Investigation, Hospital das Clínicas da Faculdade de Medicina da Universidade de São Paulo (HC-FMUSP), a university, tertiary reference center and he counseled the prophylactic treatment covering mainly tuberculosis and fungal infections.
3) The patient received three hospitalizations (COVID-19, d
engue virus infection and influenza). The claim that ustekinumab did not contribute to these hospitalizations needs to be clarified.
- A paragraph was added in the manuscript as suggested; lines 149-156
From 2017 to 2022, he was infected with COVID-19, dengue (twice), and influenza virus, and was hospitalized three times for intravenous antibiotic therapy.
The patient was infected with the dengue virus in 2018 and with dengue, COVID-19 and influenza viruses in 2022. Dengue infection is endemic in Presidente Prudente and in 2022, 17,493 individuals were infected. As with dengue, in December 2022, 1:3.2 inhabitants were infected with COVID-19 [9]. The patient’s wife and daughter were also infected with dengue and COVID-19 at the same time. He was infected with influenza virus by his parents, whom he helped care for. We suggest that treatment with ustekinumab is not responsible for the infection caused by these viruses.
4) Line 127: Loading dose of what? This needs to be specified
- Sorry for the incomplete phrase. We corrected as recommended. Lines 144-146
The 390-mg intravenous loading dose of ustekinumab was given for Crohn disease followed by maintenance with a 90 mg every 8-week;
5) Ustekinumab has immunosuppressive effects, but it is not more potent than other biolotechnology drugs used for psoriasis/psoriatic arthritis, and it is not considered a particularly potent immunosuppressant. Therefore, the term "strong" should be eliminated.
- Thank you for the observation, the text was amended as suggested.
6) It should be specified that inhibition of IL-23 acts on Th17 cells.
- A paragraph was added to the manuscript as suggested: Lines 252-258
IL-23 is a pro-inflammatory cytokine that causes differentiation and activation of Th17 cells. IL-23, together with TNF-α, supports Th17 cell development, and cytokines produced by Th1 and Th17 cell populations play a key role in the development and maintenance of psoriatic lesions and in activating the immune system. Ustekinumab can inhibit IL-12 and IL-23 signaling, activation, and cytokine production, resulting in downregulation of the immune system, which reduces inflammation and alters the body's immune response, increasing vulnerability to infectious diseases [6].
7) Line 245: Ustekinumab is approved for the treatment of Cronn's disease.
- The correction was made and the manuscript was submitted to English proofreading company.

Reviewer 2 Report
MAJOR COMMENTS:
1. The section titled "Genetic diagnosis" is better called "Genetic testing" as the molecular analysis did not result in a diagnosis, and the patient's immunodeficiency is not explained by an underlying genetic alteration. Along these lines, line 198 indicating that the patient is "genetically immunosuppressed" is not accurate. Also, I found it surprising that the Discussion section did not get into the details of the lack of the genetic diagnosis in the patient, given the extensive family history suggesting an inherited combined immunodeficiency phenotype. While lines 227-229 give a general take on precision medicine, the implication of the inconclusive genetic testing also warrants discussion, including any future directions in expanding the study or re-analyzing sequencing results.
2. Since the main goal of the study is to evaluate the impact of ustekinumab therapy on increasing the susceptibility to infections in an immunosuppressed individual, it might be useful to discuss the content of Figure 3 more elaborately in the Discussion section, which can help understand how the patient's infection spectrum changed before and after treatment.
3. Please check -- A lot of the language is repeated verbatim between the case report and the discussion sections.
MINOR COMMENTS:
1. Abstract: Lines 16-17 and lines 22-23 need to be revised for sense.
2. Reference #1 may not be an appropriate citation for lines 37-38 or 44-45.
3. Reference #2 and #3 do not sufficiently explain lines 39-40 and 53-54, respectively.
4. Line 49 - Should this be "Autoimmune diseases accounted for about 50.0% of patients diagnosed with IEI"?
5. Line 61-62 - Please provide reference(s) for "The role of ustekinumab in triggering infections in immunocompetent patients has been investigated"
6. Line 69 states that the patient was 32yo in 2016; however, in line 77, it says he was 16yo in 2006. This has to be corrected to reflect the correct age in both instances.
7. Fig 2: it might help to orient the reader by timestamping the images to show the improvement across time.
8. Lines 127-128: It is not explicitly stated what IV antibiotic therapy was administered, and the 90-mg IV loading dose is not explicitly stated to be in reference to Ustekinumab.
9. Lines 195-196: Should this line read "a broad literature was available on its effects in triggering infectious diseases in healthy individuals; however…"
10. Line 224-227: This sentence can be edited for sense
11. Line 243-245: This sentence can be edited for readability
Regarding self-citations, I found that reference #1 is a paper by the authors as well, but not appropriate in both instances cited in this manuscript.
There are some minor language edits required such as missing word or subject-verb agreement, which can be fixed by another review of the manuscript by the authors.
Round 2
Reviewer 1 Report
The authors responded appropriately to my comments